# Correlation between Cumulative Methotrexate Dose, Metabolic Syndrome and Hepatic Fibrosis Detected by FibroScan in Rheumatoid Arthritis Patients

**DOI:** 10.3390/medicina59061029

**Published:** 2023-05-26

**Authors:** Ratchaya Lertnawapan, Soonthorn Chonprasertsuk, Sith Siramolpiwat, Kanon Jatuworapruk

**Affiliations:** 1Division of Allergy, Immunology and Rheumatology, Department of Internal Medicine, Faculty of Medicine, Thammasat University, Pathumthani 12120, Thailand; ratchayas@yahoo.com; 2Division of Gastroenterology and Hepatology, Department of Internal Medicine, Faculty of Medicine, Thammasat University, Pathumthani 12120, Thailandsithsira@gmail.com (S.S.)

**Keywords:** rheumatoid arthritis, methotrexate, DMARDs, hepatic fibrosis, metabolic syndrome

## Abstract

*Background and Objectives*: Methotrexate (MTX) is routinely prescribed for rheumatoid arthritis (RA) patients, but high cumulative doses may lead to hepatic fibrosis. Additionally, a high proportion of RA patients suffer from metabolic syndrome, which also increases the risk of hepatic fibrosis. This cross-sectional study aimed to explore the association between a cumulative MTX dose, metabolic syndrome, and hepatic fibrosis in patients diagnosed with RA. *Materials and Methods*: RA patients undergoing treatment with MTX were examined using transient elastography (TE). All patients, regardless of having hepatic fibrosis, were compared to identify the risk factors. *Results*: Two hundred and ninety-five rheumatoid arthritis patients were examined using FibroScan. One hundred and seven patients (36.27%) were found to have hepatic fibrosis (TE > 7 kPa). After multivariate analysis, only BMI (OR = 14.73; 95% CI 2.90–74.79; *p* = 0.001), insulin resistance (OR = 312.07; 95% CI 6.19–15732.13; *p* = 0.04), and cumulative MTX dosage (OR 1.03; 95% CI 1.01–1.10; *p* = 0.002) were associated with hepatic fibrosis. *Conclusions*: While the cumulative MTX dose and metabolic syndrome are both the risk factors of hepatic fibrosis, metabolic syndrome, including a high BMI and insulin resistance, poses a greater risk. Therefore, MTX-prescribed RA patients with metabolic syndrome factors should be attentively monitored for signs of liver fibrosis.

## 1. Introduction

Rheumatoid arthritis (RA) is a chronic inflammatory arthritis that attacks and destroys synovial joints. RA is one of the most common types of arthritis. Early intervention to reduce inflammation can minimize joint damage. Disease-modifying antirheumatic drugs (DMARDs) are the principal treatment to reduce the progression and symptoms of RA.

Methotrexate (MTX) is considered the first-line treatment for RA and is recommended by both the European League Against Rheumatism (EULAR) and the American College of Rheumatology (ACR) [1,2]. It is prescribed as a backbone of DMARDs for RA [3]. MTX reduces inflammation by increasing adenosine. It also suppresses dihydrofolate reductase (DHFR) and prevents purine and pyrimidine synthesis, generates reactive oxygen species (ROS), decreases adhesion molecules, alters cytokine profiles, and inhibits polyamine [4,5].

MTX can produce side effects that should be monitored; among them are steatohepatitis, hepatic fibrosis, and cirrhosis [6]. Long-term use and cumulative treatment using MTX can lead to impaired liver function [1,7,8]. Hepatic fibrosis occurs in approximately 3–5% of patients who use long-term MTX for arthritis [9,10]. Hepatic fibrosis may not be detectable by a liver function test, synthetic function test, or ultrasound [5,6,8,9,10,11,12], although hepatic fibrosis measured using FibroScan has been associated with cumulative MTX doses [8,12].

Metabolic syndromes and obesity are known risk factors for hepatic fibrosis [12]. The proportion of metabolic syndrome is higher within RA patients than it is among the overall population. RA patients with metabolic syndromes, including a high BMI and fatty liver, have a higher risk of fibrosis from MTX than RA patients who do not have metabolic syndrome [8]. In this study, we measured hepatic fibrosis in RA patients who received MTX with non-invasive transient elastography (TE) and attempted to explore the association between metabolic syndrome parameters, cumulative MTX dose, and hepatic fibrosis.

## 2. Methods

### 2.1. Study Population

This was a cross-sectional study of RA patients who had received at least 1000 mg cumulative dose of MTX. There were 1288 rheumatoid arthritis patients during the study period. Nine hundred and ninety-three patients who did not meet the inclusion criteria, had the exclusion criteria, or rejected the participation were excluded from the study. The 295 patients were enrolled at Thammasat University Hospital, Thailand between November 2020 and November 2021. The diagnosis was based on ACR criteria, 2010 [13], and the treatment adhered strictly to the current ACR and EULAR RA guidelines [1,2].

The exclusion criteria were age less than 18 years; severe comorbidities, such as chronic liver, heart, or kidney disease; aspartate aminotransferase (AST) or alanine aminotransferase (ALT) levels two times greater than ULN; current hepatotoxic drug use besides MTX and other DMARDs; alcohol consumption over 7 units per week; and contraindications such as pregnancy (Figure 1).

### 2.2. Demographics and Clinical Parameters

The patient information collected included demographics, comorbidities, RA and MTX factors, concurrent medications, and laboratory parameters. Metabolic syndrome parameters included the waist, body mass index (BMI), impaired fasting glucose (IFG), triglyceride, and high-density lipoprotein HDL. The blood samples were tested (i.e., aspartate aminotransferase (AST), alanine aminotransferase (ALT), fasting plasma glucose, etc.). The ultrasound was conducted to rule out chronic liver disease and for more information on liver imaging morphology (i.e., fatty liver) (Table 1).

### 2.3. Transient Elastography (TE)

A 3.5 MHz ultrasound transducer probe (FibroScan, EchoSens, Paris, France) was used to measure liver stiffness. Patients were fasting (nil per os; NPO) for at least 3 h prior to the study. The shear wave velocity was measured ten times in each 6 cm thick portion of the right liver lobe, 1 cm in diameter and 2–4 cm long. The average, with an interquartile range of less than 20% of the median value, was used to calculate the stiffness. Any value greater than or equal to 7 kilopascals indicated fibrosis [14]. The controlled attenuation parameter (CAP)—a measure of fat storage in the liver—was also measured by FibroScan and a value of 248 or more was considered excessive.

### 2.4. Metabolic Syndrome

The patients were diagnosed with metabolic syndrome according to the modified National Cholesterol Education Program Adult Treatment Panel III (NCEPT ATP III criteria) [15], a variation of the modified NCEP ATP III criteria. According to the modified NCEP ATP III criteria [16], the presence of at least three of the following five factors is needed for diagnosing metabolic syndrome: abdominal obesity (waist circumference > 102 cm (40 inches) in men or >88 cm (35 inches) in women); hypertriglyceridemia (triglycerides > 1.7 mmol/L); low HDL cholesterol (HDL < 1.03 mmol/L for men and <1.29 mmol/L for women); elevated blood pressure (>85 mmHg) or current use of antihypertensive medications; and impaired fasting glucose (fasting plasma glucose > 5.6 mmol/L). Specific waist circumference cut-off points were used appropriately for Asians: 90 cm for men and 80 cm for women [15].

### 2.5. Statistical Analysis

Descriptive statistics were used to describe the baseline characteristics of the cohort. Comparison between people with and without hepatic fibrosis was performed using Student’s *t*-test or X^2^ test, depending on the type of data.

A univariate analysis was performed using Stata Package (Stat Statistical Software, College Station, TX, USA), including all collected parameters as independent variables and hepatic fibrosis as the dependent variable (primary outcome). All variables with *p*-value < 0.10 were selected for inclusion in the subsequent multivariate logistic regression, using the stepwise approach. We reported the odds ratios (OR), their 95% confidence interval (CI), and the *p*-values for all the variables. The variables with *p*-value < 0.05 in the multivariate logistic regression analysis would be considered independently associated with hepatic fibrosis.

### 2.6. Ethics Approval

The study protocol was reviewed and approved by the human research ethics committee of Thammasat University (Medicine), in compliance with the declaration of Helsinki (protocol code MTU-EC-IM-6-141/60, date of approval 27 December 2017).

## 3. Results

### 3.1. Baseline Characteristics

Of the 295 RA patients enrolled in this study, 107 (36.27%) had hepatic fibrosis (Figure 1). The group of patients with fibrosis was older (*p* = 0.02) and had higher values for albumin (*p* = 0.007), AST (*p* = 0.002), ALT (*p* = 0.001), creatinine (*p* < 0.001), INR (*p* = 0.02), and uric acid (*p* < 0.001) compared to the group with no fibrosis. There was no difference in sex, alcohol consumption, hemoglobin, white blood cell count, or platelet count. The values are given in Table 1.

### 3.2. Rheumatoid Arthritis

The group of patients with hepatic fibrosis had RA longer (*p* = 0.009). Leflunomide (*p* = 0.03), a conventional synthetic DMARD, and steroids (*p* = 0.04) were prescribed more in the fibrosis group.

The fibrosis group had longer MTX use (291.63 weeks vs. 189.42 weeks, *p* = 0.001), higher dose per weight (70.46 mg/kg vs. 43.55 mg/kg, *p* < 0.001), and a higher cumulative dosage (4661.4 mg vs. 2212.07 mg, *p* < 0.001). However, both groups had similar doses per week and doses per week per weight.

### 3.3. Metabolic Syndrome

Body weight (*p* < 0.001), waist (*p* < 0.001), hip (*p* < 0.001), body mass index (BMI) (*p* < 0.001), diabetes (*p* < 0.001), impaired fasting blood glucose (IFG) (*p* < 0.001), insulin resistance (*p* < 0.001), hypertension (<0.001), fatty liver (*p* < 0.001), hyperlipidemia (HLP) (*p* < 0.001), and metabolic syndrome (*p* < 0.001) were notably higher in the hepatic fibrosis group.

The patients with hepatic fibrosis had a greater prevalence of statin prescription (*p* < 0.001). The fibrosis group had higher laboratory values of fasting blood sugar (*p* < 0.001), HbA1c (*p* < 0.001), cholesterol (*p* < 0.001), triglyceride (*p* < 0.001), HDL (*p* < 0.001), LDL (*p* < 0.001), fatty liver by sonography (*p* < 0.001), and controlled attenuation parameter (CAP) (*p* < 0.001).

### 3.4. Factors Associated with Hepatic Fibrosis

The factors significantly associated with hepatic fibrosis according to univariate analysis are listed in Table 2. These variables were subsequently included in the multivariate analysis.

Multivariate regression analysis found that BMI (OR = 14.73; 95% CI 2.90–74.79; *p* = 0.001); insulin resistance (OR = 312.07; 95% CI 6.19–15732.13; *p* = 0.04); and cumulative MTX dose (OR 1.03; 95% CI 1.01–1.10; *p* = 0.002) were significantly associated with hepatic fibrosis. (Table 3)

## 4. Discussion

In our study, we compared RA patients receiving MTX who had hepatic fibrosis to those who did not. The patients with fibrosis were significantly older and their levels of albumin, AST, ALT, creatinine, and INR were significantly higher. The fibrosis patients had a longer duration of RA, longer duration of MTX, higher cumulative MTX, higher average MTX dose per weight, more prescriptions of leflunomide or prednisolone, and higher ESR levels. In our study, multivariate regression analysis indicated that a high BMI, insulin resistance, and cumulative MTX dose were associated with hepatic fibrosis.

The main treatment for RA, recommended by both ACR and EULAR, is MTX [1,2]. Long-term use and a high cumulative dose of MTX are considered risk factors for hepatic fibrosis [12,17,18,19,20,21,22,23,24,25]. However, some studies found that neither the duration nor dose of MTX correlated with liver disease [22,23,24,25]. Our study found that a cumulative dosage of MTX was associated with hepatic fibrosis, which is concordant with the pathogenesis of MTX in liver toxicity.

Meta-analysis suggests that MTX could cause transient elevations of ALT, and at least one study found somewhat elevated ALT in up to 50% of MTX users, but significant elevations are rare [6,26,27]. Folic acid has been reported to reduce ALT and lower the risk of hepatotoxicity [9,27]. However, liver enzymes and synthetic function may be normal even in severe hepatic fibrosis [5,9]. Furthermore, some studies indicate that fibrosis could result from oxidative stress [28].

Higher cumulative doses of MTX are increasingly toxic to the liver [11]. Lower doses can cause similar damage if patients have preexisting liver disease (e.g., hepatitis B and hepatitis C), or risk factors such as metabolic syndrome [5,6,12,17,26,27,29].

The most prescribed treatment for many autoimmune diseases such as inflammatory myositis, psoriasis, psoriatic arthritis, and RA is MTX. MTX can synergistically augment preexisting hepatic diseases [8]. A high cumulative MTX dose, obesity, old age, and alcohol can further increase liver damage. Smaller cumulative MTX doses in combination with metabolic syndrome or nonalcoholic steatohepatitis (NASH) can also increase the risk of fibrosis [19]. A high BMI, high cholesterol, and long duration of inflammatory arthritis can increase the risk of MTX-induced liver fibrosis [8].

We previously demonstrated that RA patients with metabolic syndrome features (high BMI, IFG, and fatty liver) carry a higher risk of developing hepatic fibrosis [8]. The current study found some association between hepatic fibrosis and body weight, high BMI, IFG, fatty liver, and statin use from univariate analysis. Only BMI and insulin resistance, however, were retained in significant association with hepatic fibrosis after multivariate analysis.

RA patients have more metabolic syndrome, obesity, dyslipidemia, diabetes, and cardiovascular disorders than the general population [19]. RA patients with NASH or metabolic syndrome should be closely monitored during MTX use [8]. Although AST and ALT were not statistically associated with liver fibrosis, they may remain clinically valuable due to their low price and wide availability. Advice on lifestyle modification and weight reduction should be provided for all RA patients with metabolic syndrome taking MTX. The risk of liver fibrosis in patients with RA has been found to be more closely associated with metabolic syndrome than with a cumulative MTX dose [30,31]. Our current study also found that hepatic fibrosis was more significantly associated with metabolic syndrome than with cumulative MTX dose.

This study has some limitations. Our study results may have limited applicability due to the cross-sectional, single-center design involving a homogeneous group of participants (ethnic Thai). We did not categorize liver fibrosis by severity due to the lack of a standardized threshold for liver elasticity in Asian populations taking MTX. Another limitation is that we did not include a histopathological examination and a Child–Pugh score assessment for our study cohort.

In conclusion, a cumulative MTX dose is a contributing risk factor for hepatic fibrosis, especially when the patients have metabolic syndrome. In particular, the metabolic syndrome components, including BMI and insulin resistance, significantly correlated with hepatic fibrosis in MTX-prescribed RA patients. These patients are at high propensity and should be attentively monitored by a physician. A hepatologist consultation should be considered in complicated cases.

## Figures and Tables

**Figure 1 medicina-59-01029-f001:**
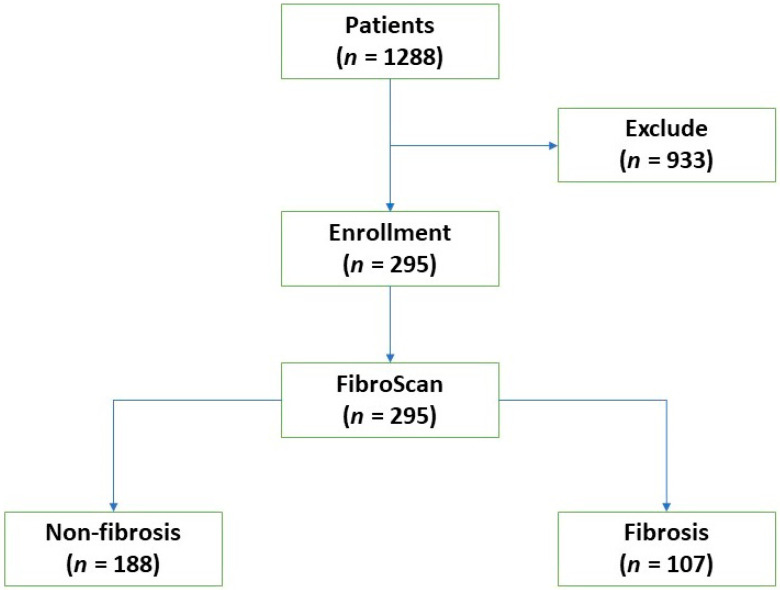
Study flowchart.

**Table 1 medicina-59-01029-t001:** Baseline characteristics of the study population (N = 295).

Parameters		Non-Fibrosis (N = 188)	Fibrosis (N = 107)	*p*-Value
	Sex (female) (%)	154 (81.91)	87 (81.31)	1
	Age (years)	55.36 ± 15.12	59.49 ± 11.46	0.02
	BWt (kg)	52.37 ± 7.71	66.58 ± 1.00	<0.001
	Height (cm)	157.77 ± 7.05	156.85 ± 8.94	0.33
	Waist (cm)	72.15 ± 6.06	86.24 ± 8.23	<0.001
	Hip (cm)	91.14 ± 5.76	95.89 ± 7.51	<0.001
	Waist–hip ratio (WHR)	1.21 ± 5.78	0.89 ± 0.05	0.58
	BMI (kg/m^2^)	20.99 ± 2.38	26.98 ± 2.59	<0.001
Comorbidities	Alcohol consumption (%)	2 (1.06)	3 (2.80)	0.36
	DM (%)	13 (6.9)	30 (28.04)	<0.001
	IFG (%)	18 (9.57)	64 (59.81)	<0.001
	Insulin resistance (%)	31 (16.49)	94 (87.85)	<0.001
	HT (%)	52 (27.66)	74 (69.16)	<0.001
	Fatty liver (%)	10 (5.32)	53 (49.53)	<0.001
	HLP (%)	53 (28.19)	102 (95.33)	<0.001
	Metabolic syndrome (%)	3 (1.60)	107 (100)	<0.001
	HBV/HCV (%)	0 (0)	0 (0)	
RA and MTX factors	RA duration (years)	8.47 ± 4.09	10.07 ± 5.77	0.006
	AntiCCP (%)	161 (85.64)	98 (91.59)	0.14
	Rheumatoid factor (%)	143 (76.06)	88 (82.24)	0.24
	MTX duration (weeks)	189.42 ± 136.65	291.63 ± 168.23	0.001
	MTX average (mg/w)	10.51 ± 3.01	11.07 ± 3.38	0.39
	MTX cumulative(mg)	2212.07 ± 1021.32	4661.4 ± 3192.51	<0.001
	MTX per weight(mg/kg)	43.55 ± 22.35	70.46 ± 44.98	<0.001
	MTX average/wt (mg/w/kg)	0.20 ± 0.07	0.18 ± 0.06	0.11
Concurrent medications	Other DMARDs (%)	161 (85.64)	98 (91.59)	0.14
	Hydroxychloroquine (%)	51 (27.13)	34 (31.78)	0.42
	Sulfasalazine (%)	75 (39.89)	34 (31.78)	0.17
	Leflunomide (%)	104 (55.32)	78 (72.90)	0.003
	Azathioprine (%)	6 (3.19)	5 (4.67)	0.54
	Biologic agent (%)	17 (9.04)	11 (10.28)	0.84
	Steroid (%)	56 (29.79)	45 (42.06)	0.04
	Prednisolone dosage (mg/d)	0.68 ± 1.51	0.98 ± 1.87	0.14
	NSAIDs (%)	39 (20.75)	20 (18.69)	0.84
	Folic (%)	188 (100)	107 (100)	
	Statin (%)	40 (21.28)	62 (57.94)	<0.001
Laboratory parameters	ESR (mm/h)	37.45 ± 21.21	42.90 ± 21.21	0.03
	CRP (mg/L)	9.20 ± 20.03	7.75 ± 11.78	0.49
	Hemoglobin (g/dL)	11.75 ± 1.23	11.96 ± 1.40	0.35
	WBC (×10^9^/L)	6478.56 ± 2279.88	6451.50 ± 1847.04	0.92
	Platelet (×10^9^/L)	272,262.20 ± 77,879.23	269,411.20 ± 74,477.29	0.76
	FBS (mg/dL)	93.43 ± 14.10	110.06 ± 16.19	<0.001
	HbA1c (%)	5.44 ± 0.87	6.05 ± 0.57	<0.001
	Albumin (g/dL)	3.75 ± 0.47	3.89 ± 0.40	0.007
	AST (IU/L)	23.28 ± 5.92	26.94 ± 10.69	0.002
	ALT (IU/L)	20.56 ± 9.93	27.04 ± 17.51	0.001
	Creatinine (mg/dL)	0.74 ± 0.17	0.81 ± 0.20	<0.001
	INR	1.00 ± 0.06	1.01 ± 0.07	0.02
	Cholesterol (mg/dL)	181.38 ± 30.27	229.79 ± 39.07	<0.001
	Triglyceride (mg/dL)	88.36 ± 42.29	148.17 ± 58.94	<0.001
	HDL (mg/dL)	62.25 ± 16.09	52.80 ± 15.63	<0.001
	LDL (mg/dL)	101.95 ± 24.46	137.20 ± 28.87	<0.001
	Uric acid (mg/dL)	4.55 ± 0.71	5.12 ± 0.92	<0.001
	Abnormal ultrasound (fatty liver by ultrasound) (%)	10 (5.32)	53 (49.53)	<0.001
	CAP	195.24 ± 30.47	249.43 ± 43.50	<0.001
	FibroScan (kPa)	4.94 ± 0.89	9.09 ± 3.31	<0.001

WHR: waist–hip ratio; BMI: body mass index; DM: diabetes mellitus; IFG: impaired fasting blood glucose; HT: hypertension; HLP: hyperlipidemia; HBV: hepatitis B virus; HCV: hepatitis C virus; RA: rheumatoid arthritis; AntiCCP: anti-cyclic citrullinated peptide; MTX: methotrexate; DMARDs: disease modifying antirheumatic drug; ESR: erythrocyte sedimentation rate; CRP: C-reactive protein; WBC: white blood cell; FBS: fasting blood sugar; HbA1c: hemoglobin A1c; AST: aspartate transaminase; ALT: alanine transaminase; INR: international normalized ratio; HDL: high-density lipoprotein; LDL: low-density lipoprotein; CAP: controlled attenuation parameter.

**Table 2 medicina-59-01029-t002:** The univariate logistic regression analysis of factors associated with significant hepatic fibrosis.

	Odds Ratio	95% Confidence Interval	*p*-Value
Age	1.02	1.00–1.04	0.016
BMI (kg/m^2^)	8.29	4.46–15.42	<0.001
Waist (cm)	1.32	1.24–1.40	<0.001
Hip (cm)	1.12	1.08–1.17	<0.001
DM	5.24	2.59–10.60	<0.001
IFG	14.06	7.56–26.15	<0.001
Insulin resistance	36.62	18.25–73.47	<0.001
HT	5.86	3.49–9.87	<0.001
Fatty liver	17.47	8.33–36.66	<0.001
HLP	51.96	20.05–134.67	<0.001
RA duration (y)	1.07	1.02–1.13	0.008
MTX duration (w)	1.00	1.00–1.01	0.003
MTX cumulative dose (mg)	2.01	1.02–2.13	<0.001
MTX/kg (mg/kg)	1.03	1.02–1.05	<0.001
Leflunomide	2.17	1.30–3.63	0.003
Steroid	1.71	1.04–2.81	0.03
FBS	1.08	1.06–1.11	<0.001
HbA1c	13.33	6.82–26.04	<0.001
Albumin (g/dL)	2.20	1.22–3.96	0.008
AST (IU/L)	1.07	1.03–1.11	<0.001
ALT (IU/L)	1.04	1.02–1.07	<0.001
Creatinine (mg/dL)	9.30	2.41–35.94	0.001
INR	79.67	2.06–3081.80	0.02
Cholesterol (mg/dL)	1.04	1.03–1.05	<0.001
Triglyceride (mg/dL)	1.03	1.02–1.03	<0.001
HDL (mg/dL)	0.95	0.94–0.97	<0.001
LDL (mg/dL)	1.05	1.04–1.06	<0.001
Uric (mg/dL)	2.58	1.81–3.67	<0.001
Fatty liver by ultrasound	19.52	8.74–43.60	<0.001

BMI: body mass index; DM: diabetes mellitus; IFG: impaired fasting blood glucose; HT: hypertension; HLP: hyperlipidemia; RA: rheumatoid arthritis; MTX: methotrexate; FBS: fasting blood sugar; HbA1c: hemoglobin A1c; AST: aspartate transaminase; ALT: alanine transaminase; INR: international normalized ratio; HDL: high-density lipoprotein; LDL: low-density lipoprotein.

**Table 3 medicina-59-01029-t003:** The multivariate logistic regression analysis of the factors associated with significant hepatic fibrosis.

	ODDS Ratio	95% Confidence Interval	*p*-Value
BMI (kg/m^2^)	14.73	2.90–74.79	0.001
Insulin resistance	312.07	6.19–15,732.13	0.004
HT	5.35	0.29–99.52	0.26
HLP	5.50	0.34–89.45	0.23
MTX cumulative dose (mg)	1.03	1.01–1.10	0.002

BMI: body mass index; HT: hypertension; HLP: hyperlipidemia; MTX: methotrexate.

## Data Availability

Data is contained within the article.

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
