# Peer review of "Correlation between Cumulative Methotrexate Dose, Metabolic Syndrome and Hepatic Fibrosis Detected by FibroScan in Rheumatoid Arthritis Patients"

_medicina, 2023, doi:10.3390/medicina59061029_

Round 1

Reviewer 1 Report

This work puts the attention on the methotrexate-induced risk of liver fibrosis in patient with rheumatoid arthritis. The topic is of particular interest because, although the availability of new drugs, MTX remains a cornerstone for RA treatment. The research is well designed and the article is well written but there are some minor correction that I suggest to do to improve its quality:

Line 58: authors should be more cautious in this statement: I suggest to write "MTX CAN produce side effects..."

Lines 60-61: I checked the references but I cannot find the indicated 4% incidence of hepatic fibrosis in patients using MTX. Please provide more accurate information on where this value comes from.  

Lines 68-69:  I think that the mention of psoriatic patients is out of topic because psoriasis is a disease with different pathogenetic pathways in comparison to RA

Table 1,2,3 : in descriptive statistics, continuous variables should be represented as means and standard deviation. Here only mean is indicated 

Table 1 and "results"section: authors highlighted  the significancy of the difference in term of labs parameters (ALT, AST, creatinine...) between patients with and without liver fibrosis. Anyway, the clinical significance of this difference seems to be negligible. I suggest to point out this or to evidence only the clinically significant differences. 

Lines 171-179: results of univariate analysis is also reported in table 2. I think the detailed description in the text is redundant and can be avoided to make the reading easier. 

Lines 222-230: as above, I suggest to cut down the discussion on psoriasis because it is off topic

Author Response

Response to reviewers

ID: medicina-2402627

We thank the reviewers for their comments. Our point-by-point responses are listed below.

Reviewer 1:

This work puts the attention on the methotrexate-induced risk of liver fibrosis in patient with rheumatoid arthritis. The topic is of particular interest because, although the availability of new drugs, MTX remains a cornerstone for RA treatment. The research is well designed and the article is well written but there is some minor correction that I suggest to do to improve its quality:

Line 58: authors should be more cautious in this statement: I suggest to write "MTX CAN produce side effects..."

Response: The phrase has been corrected (line 58).

Lines 60-61: I checked the references but I cannot find the indicated 4% incidence of hepatic fibrosis in patients using MTX. Please provide more accurate information on where this value comes from. 

Response: Thank you for pointing this out. We have corrected the number to 3-5% (the incidence was 3% in reference 9 and 5% in reference 10). (line 60-61)

Lines 68-69:  I think that the mention of psoriatic patients is out of topic because psoriasis is a disease with different pathogenetic pathways in comparison to RA

Response: The phrase has been removed as suggested. (line 68-69)

Table 1,2,3: in descriptive statistics, continuous variables should be represented as means and standard deviation. Here only mean is indicated

Response: stand deviations are now included for all reports of continuous variables (table 1). Table 2 and 3 do not contain continuous variable so no adjustment was made to them.

Table 1 and "results" section: authors highlighted the significance of the difference in term of labs parameters (ALT, AST, creatinine...) between patients with and without liver fibrosis. Anyway, the clinical significance of this difference seems to be negligible. I suggest to point out this or to evidence only the clinically significant differences.

Response: The results section has been revised, with statements on non-significant variables removed. (line 151-169)

Lines 171-179: results of univariate analysis is also reported in table 2. I think the detailed description in the text is redundant and can be avoided to make the reading easier.

Response: The excess description has been removed from the results section (line 173-180) and the discussion section (line 201-206).

Lines 222-230: as above, I suggest to cut down the discussion on psoriasis because it is off topic

Response: The section on psoriatic arthritis has been removed (224-233).

Reviewer 2 Report

The authors present a manuscript on the hepatotoxic (liver fibrosis) effects of methotrexate (MTX) use in patients with rheumatoid arthritis. The study is interesting and well conducted. However, it needs some clarifications:

1)      The extent of liver elasticity reduction obtained by fibroscan is not indicated. Authors should specify whether there was mild, moderate, or severe fibrosis and possibly indicate the corresponding METAVIR values as obtainable from liver biopsy

2)      Although the authors have cited studies that dissociate aminotransferase values with the extent of fibrosis, assessment of ALT and AST is an easily obtainable parameter during MTX therapy. The authors should comment on the clinical value of such monitoring, which is certainly easier to obtain than the use of fibroscan.

3)      The finding that patients with fibrosis had lower than normal albumin and INR values suggests the presence of liver cirrhosis. An evaluation by Child-Pugh score is helpful in understanding the clinical impact of MTX-induced fibrosis.

4)      The key role of BMI suggests that patients should be advised to follow a diet to minimize the side effects of MTX. A comment on this is helpful.

Enghlish is good and needs only minor editing

Author Response

Response to reviewers

ID: medicina-2402627

We thank the reviewers for their comments. Our point-by-point responses are listed below.

Reviewer 2:

The authors present a manuscript on the hepatotoxic (liver fibrosis) effects of methotrexate (MTX) use in patients with rheumatoid arthritis. The study is interesting and well conducted. However, it needs some clarifications:

1) The extent of liver elasticity reduction obtained by fibroscan is not indicated. Authors should specify whether there was mild, moderate, or severe fibrosis and possibly indicate the corresponding METAVIR values as obtainable from liver biopsy

Response: Thank you for the comment. When designing the study, we did not categorize the extent of liver elasticity according to degree of severity due to lack of standardized threshold for liver fibrosis caused by MTX, especially in Asian people. We therefore decided to adhere to protocols and threshold successfully used by previous similar research (Park SH, et al. Joint Bone Spine. 2010;77(6):588-92.) in people using MTX. We have added a statement on this limitation to our discussion section (line 262-263).

2) Although the authors have cited studies that dissociate aminotransferase values with the extent of fibrosis, assessment of ALT and AST is an easily obtainable parameter during MTX therapy. The authors should comment on the clinical value of such monitoring, which is certainly easier to obtain than the use of fibroscan.

Response: We agree that AST/ALT remains a valuable test in clinical practice despite the lack of statistically significant association from studies. We have therefore added a statement on this issue into our discussion section (line 250-252).

3) The finding that patients with fibrosis had lower than normal albumin and INR values suggests the presence of liver cirrhosis. An evaluation by Child-Pugh score is helpful in understanding the clinical impact of MTX-induced fibrosis.

Response: Thank you for the comment. We believe Child-Pugh score is most meaningful when used for assessment of people with established cirrhosis. We therefore did not assess Child-Pugh score in our patients since they would have no previous diagnosis of cirrhosis. However, we agree that measurement of Child-Pugh score would have increased the interpretability of our study results. We have added this limitation to the discussion section (line 264-265).

4) The key role of BMI suggests that patients should be advised to follow a diet to minimize the side effects of MTX. A comment on this is helpful.

Response: Thank you for the comment. We have added a statement on weight reduction into our discussion section (line 252-254).